# More Is Less: Learning Efficient Video Representations by Big-Little Network and Depthwise Temporal Aggregation

**Quanfu Fan**[†,1]**, Chun-Fu (Richard) Chen**[†,2]**, Hilde Kuehne**[1]**, Marco Pistoia**[2]**, David Cox**[1]

†: Equal contribution

[1]MIT-IBM Waston AI Lab, Cambridge, MA 02142, USA

[2]IBM T.J. Waston Research Center, Yorktown Heights, NY 10598, USA

`{qfan, chenrich, pistoia}@us.ibm.com`, `{kuehne, david.d.cox}@ibm.com`

## Abstract

Current state-of-the-art models for video action recognition are mostly based on expensive 3D ConvNets. This results in a need for large GPU clusters to train and evaluate such architectures. To address this problem, we present an lightweight and memory-friendly architecture for action recognition that performs on par with or better than current architectures by using only a fraction of resources. The proposed architecture is based on a combination of a deep subnet operating on low-resolution frames with a compact subnet operating on high-resolution frames, allowing for high efficiency and accuracy at the same time. We demonstrate that our approach achieves a reduction by $3 \sim 4$ times in FLOPs and $\sim 2$ times in memory usage compared to the baseline. This enables training deeper models with more input frames under the same computational budget. To further obviate the need for large-scale 3D convolutions, a temporal aggregation module is proposed to model temporal dependencies in a video at very small additional computational costs. Our models achieve strong performance on several action recognition benchmarks including Kinetics, Something-Something and Moments-in-time. The code and models are available at `https://github.com/IBM/bLVNet-TAM`.

## 1 Introduction

Current state-of-the-art approaches for video action recognition are based on convolutional neural networks (CNNs). These include the best performing 3D models, such as I3D [1] and ResNet3D [2], and some effective 2D models, such as Temporal Relation Networks (TRN) [3] and Temporal Shift Modules (TSM) [4]. A CNN-based model usually considers a sequence of frames as input, obtained through either uniform or dense sampling from a video [1, 5]. In general, Longer input sequences yield better recognition results. However, one problem arising for a model requesting more input frames is that the GPU resources required for training and inference also significantly increase in both memory and time. For example, the top-performing I3D models [1] on the Kinetics [6] dataset were trained with 64 frames on a cluster of 32 GPUs, and the non-local network [7] even uses 128 frames as input. Another problem for action recognition is the lack of effective methods for temporal modeling when moving away from 3D spatiotemporal convolutions. While 2D convolutional models are more resource-friendly than their 3D counterparts, they lack expressiveness over time and thus cannot take much benefit from richer input data.

In this paper, we present an efficient and memory-friendly spatio-temporal representation for action recognition, which enables training of deeper models while allowing for more input frames. The first part of our approach is inspired by the Big-Little-Net architecture (*bLNet* [8]). We propose a new

video architecture that has two network branches with different complexities: one branch processing low-resolution frames in a very deep subnet, and another branch processing high-resolution frames in a compact subnet. The two branches complement each other through merging at the end of each network layer. With such a design, our approach can process twice as many frames as the baseline model without compromising efficiency. We refer to this architecture as "Big-Little-Video-Net" (*bLVNet*).

In light of the limited ability of capturing temporal dependencies in *bLVNet*, we further develop an effective method to exploit temporal relations across frames by a so called "Depthwise Temporal Aggregation Module" (*TAM*). The method enables the exchange of temporal information between frames by weighted channel-wise aggregation. This aggregation is made learnable with $1 \times 1$ depthwise convolution, and implemented as an independent network module. The temporal aggregation module can be easily integrated into the proposed network architecture to progressively learn spatio-temporal patterns in a hierarchical way. Moreover, the module is extremely compact and adds only negligible computational costs and parameters to *bLVNet*.

Our main contributions lie in the following two interconnected aspects: (1) We propose a lightweight video architecture based on dual-path network to learn video features, and (2) we develop a temporal aggregation module to enable effective temporal modeling without the need for computationally expensive 3D convolutions.

We evaluate our approach on the Kinetics-400 [6], Something-Something [9] and Moments-in-time [10] datasets. The evaluation shows that *bLVNet-TAM* successfully allows us to train action-classification models with deeper backbones (i.e., ResNet-101) as well as more (up to 64) input frames, using a single compute node with 8 Tesla V100 GPUs. Our comprehensive experiments demonstrate that our approach achieves highly competitive results on all datasets while maintaining efficiency. Especially, it establishes a new state-of-the-art result on Something-Something and Moments-in-time by outperforming previous approaches in the literature by a large margin.

## 2 Related Work

Activity classification has always been a challenging research topic, with first attempts reaching back by almost two decades [11]; deep-learning architectures nowadays achieve tremendous recognition rates on various challenging tasks, such as Kinetics [1], ActivityNet [12], or Thumos [13].

Most successful architectures in the field are usually based on the so-called two-stream model [14], processing a single RGB frame and optical-flow input in two separate CNNs with a late fusion in the upper layers. Over the last years, many approaches extend this idea by processing a stack of input frames in both streams, thus extending the temporal window of the architecture form 1 to up to 128 input frames per stream. To further capture the temporal correlation in the input over time, those architectures usually make use of 3D convolutions as, e.g., in I3D [1], S3D [15], and ResNet3D [2], usually leading to a large-scale parameter space to train.

Another way to capture temporal relations has been proposed by [5], [3], and [4]. Those architectures mainly build on the idea of processing videos in the form of multiple segments, and then fusing them at the higher layers of the networks. The first approach with this pattern was the so-called Temporal Segment Networks (TSN) proposed by Wang *et al.* [5]. The idea of TSN has been extended by Temporal Relation Networks (TRN) [3], which apply the idea of relational networks to the modeling of temporal relations between observations in videos. Another approach for capturing temporal contexts has been proposed by Temporal Shift Modules (TSM) [4]. This approach shifts part of the channels along the temporal dimension, thereby allowing for information to be exchanged among neighboring frames. More complex approaches have been tried as well, e.g. in the context of non-local neural networks [7]. Our temporal aggregation module is based on depthwise $1 \times 1$ convolutions to capture temporal dependencies across frames effectively.

Separate convolutions are considered in approaches such as [15, 16] to reduce costly computation in 3D convolutional models. More recently, SlowFast Network [17] uses a dual-pathway network to process a video at both slow and fast frame rates. The fast pathway is made lightweight, similar to Little Net in our proposed architecture. However, our approach reduces computation based on both a lightweight architecture and low image resolution. Furthermore, the recent work Timeception [18] applies the concept of "Inception" to temporal domain for capturing long-range temporal dependencies

in a video. The Timeception layers involve group convolutions at different time scales while our TAM layers only use depthwise convolution. As a result, the Timeception has significantly more parameters than the TAM (10% vs. 0.1% of the total model parameters).

## 3   Our Approach

We aim at developing efficient and effective video representations for video understanding. To address the computational challenge imposed by the desired long input to a model, we propose a new video architecture based on the Big-Little network (*bLNet*) [8] for learning video features.We first give a brief recap of *bLNet* in Section 3.1. We then show, in Section 3.2, how to extend *bLNet* to an efficient video architecture that allows for seeing more frames with less computation and memory. An example of the proposed network architecture can be found in the supplementary material (Section A).

To make temporal modeling more effective in our approach, we further develop a temporal aggregation module (TAM) to capture short-term as well as long-term temporal dependencies across frames. Our method is implemented as a separate network module and integrated with the proposed architecture seamlessly to learn a hierarchical temporal representation for action recognition. We detail this method in Section 3.3.

### 3.1   Recap of Big-Little Network

The Big-Little Net, abbreviated as *bLNet* in [8], is a CNN architecture for learning strong feature representations by combining multi-scale image information. The *bLNet* processes an image at different resolutions using a dual-path network, but with low computational loads based on a clever design. The key idea is to have a high-complexity subnet (*Big-Net*) along with a low-cost one (*Little-Net*) operate on the low-scale and high-scale parts of an image in parallel. By such a design, the two subnets learn features complementary to each other while using less computation. The two branches are merged at the end of each network layer to fuse the low-scale and high-scale information so as to form a stronger image representation. The *bLNet* approach demonstrates improvement of model efficiency and performance on both object and speech recognition, using popular architectures such as ResNet, ResNeXt and SEResNeXt. More details on *bLNet* can be found in the original paper. In this work, we mainly adopt *bLResNet*-50 and *bLResNet*-101 as backbone for our proposed architecture.

### 3.2   Big-Little Video Network as Video Representation

We describe our architecture in the context of 2D convolutions. However our approach is not specific to 2D convolutions and potentially extendable to any architecture based on 3D convolutions.

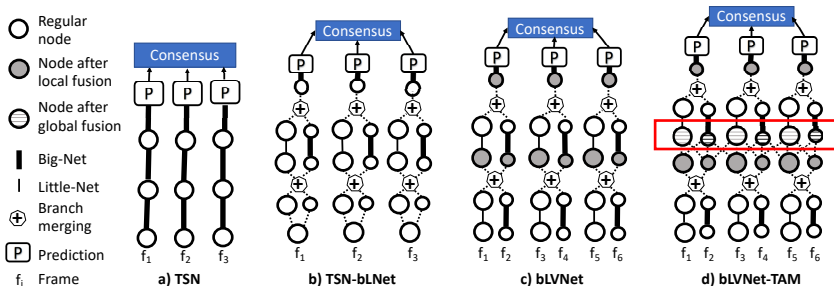

Figure 1: Different architectures for action recognition. a) **TSN** [5] uses a shared CNN to process each frame independently, so there is no temporal interaction between frames. b) **TSN-bLNet** is a variant of TSN that uses *bLNet* [8] as backbone. It is efficient, but still lacks temporal modeling. c) **bLVNet** feeds odd and even frames separately into different branches in *bLNet*. The branch merging at each layer (*local fusion*) captures short-term temporal dependencies between adjacent frames. d) **bLVNet-TAM** includes the proposed aggregation module, represented as a red box, which further empowers *bLVNet* to model long-term temporal dependencies across frames (*global fusion*).

The approach of Temporal Segment Networks (TSN) [5] provides a generic framework for learning video representations. With a shared 2D ConvNet as backbone, TSN performs frame-level predictions and then aggregates the results into a final video-level prediction (Fig. 1a)). The framework of TSN is efficient and has been successfully adopted by some recent approaches for action recognition such as TRN [3] and TSM [4]. Given its efficiency, we also choose TSN as the underlying video framework for our work.

Let $\mathbf{F} = \{f_t | t = 1 \cdots n\}$ be a set of sampled input frames from a video. We divide $\mathbf{F}$ into two groups, namely odd frames $\mathbf{F_{odd}} = \{f_k \in \mathbf{F} | \mod(k, 2) \neq 0\}$ at half of the input image resolution, and even frames $\mathbf{F_{even}} = \{f_k \in \mathbf{F} | \mod(k, 2) = 0\}$ at the input image resolution. For convenience, from now on, $\mathbf{F_{odd}}$ is referred to as *big* frames and $\mathbf{F_{even}}$ as *little* frames. Note that big branch can take either of a pair of frames as input and the other frame goes to the little branch.

In TSN, all input frames are ordered as a batch of size $n$, where the $t^{th}$ element corresponds to the $t^{th}$ frame. We denote the input and output feature maps of the $t^{th}$ frame at the $k^{th}$ layer of the model by $\mathbf{x}_t^k \in \mathbb{R}^{C \times W \times H}$ and $\mathbf{y}_t^k \in \mathbb{R}^{C \times W \times H}$, respectively. Whenever possible, we omit $k$ for clarity.

The *bLNet* can be directly plugged into TSN as the backbone network for learning video-level representation. We refer to this architecture as *TSN-bLNet* to differentiate it from the vanilla TSN (Fig. 1b)). This network fully enjoys the efficiency of *bLNet*, cutting the computational costs down by $1.6 \sim 2$ times according to [8]. Mathematically, the output $\mathbf{y}_t$ can be written as

$$\mathbf{y}_t = \mathcal{F}(net_B([\mathbf{x}_t]_{1/2}) + net_L(\mathbf{x}_t), \theta_t). \tag{1}$$

Here $[\cdot]_s$ is an operator scaling a tensor up or down by a factor of $s$ in the spatial domain; $net_B$ and $net_L$ are the Big-Net and Little-Net in the *bLNet* aforementioned; and $\theta_t$ are the model parameters. Following [8], $\mathcal{F}$ indicates an additional residual block applied after merging the big and little branches to stabilize and enhance the combined feature representation.

The architecture described above only learns features from a single frame, so there are no interactions between frames. Alternatively, we can feed the odd and even frames separately into the big and little branches so that each branch obtains complementary information from different frames. This idea is illustrated in Fig. 1c) and the output $\mathbf{y}_t$ in this case can be expressed by

$$\mathbf{y}_t = \begin{cases} \mathcal{F}(net_B(\lfloor \mathbf{x}_t \rfloor_{1/2}) + net_L(\mathbf{x}_{t+1}), \theta_t), & \text{if } \mod(t, 2) \neq 0 \\ \mathbf{y}_{t-1}, & \text{otherwise} \end{cases} \tag{2}$$

While the modification proposed above is simple, it leads to a new video architecture, which is called Big-Little-Video-Net, or *bLVNet* for short. The *bLVNet* makes two distinct differences from *TSN-bLNet*. Firstly, without increasing any computation, it can take input frames two times as many as *TSN-bLNet*. We shall demonstrate the benefit of leveraging more frames for temporal modeling in Section 4. Furthermore, the *bLVNet* has $1.5 \sim 2.0\times$ fewer FLOPs than TSN while seeing frames twice as many as TSN, thanks to the efficiency of the dual-path network. Secondly, the merging of the two branches in *bLVNet* now happens on two different frames carrying temporal information. We call this type of temporal interaction by *local fusion*, since it only captures temporal relations between two adjacent frames. In spite of that, local fusion gives rise to a significant performance boost for recognition, as shown later in Section 4.3.

### 3.3 Temporal Aggregation Module

Temporal modeling is a challenging problem for video understanding. Theoretically, adding a recurrent layer such as LSTM [19] on top of a 2D ConvNet seems like a promising means to capture temporal ordering and long-term dependencies in actions. Nonetheless, such approaches are not practically competent with 3D ConvNets [1], which use spatio-temporal filters to learn hierarchical feature representations. One issue with 3D models is that they are heavy in parameters and costly in computation, making them hard to train. Even though some approaches like S3D [15] and R(2+1)D [16] alleviates this issue by separating a 3D convolution filter into a 2D spatial component followed by a 1D temporal component, they are in general still more expensive than 2D ConvNet models.

With the efficient *bLVNet* architecture described above, our goal is to further improve its spatio-temporal representation by effective temporal modeling. The local fusion in *bLVNet* only exploits temporal relations between neighbored frames. To address this limitation, we develop a method

to capture short-term as well as long-term dependencies across frames. Our basic idea is to fuse temporal information at each time instance by weighted channel-wise aggregation. As detailed below, this idea can be efficiently implemented as a network module to progressively learn spatio-temporal patterns in a hierarchical way.

Let $\mathbf{y}_t$ be the output (i.e. neural activation) of the $t^{th}$ frame $f_t$ at a layer of the network (see Eq. 2). To model the temporal dependencies between $f_t$ and its neighbors, we aggregate the activations of all the frames within a temporal range $r$ around $f_t$. A weight is learned for each channel of the activations to indicate its relevance. Specifically, the aggregation results can be written as

$$\hat{\mathbf{y}}_t = ReLU\left(\sum_{j=-\lfloor r/2 \rfloor}^{j=\lfloor r/2 \rfloor} \mathbf{w}_j \otimes \mathbf{y}_{t+j}\right), \tag{3}$$

where $\otimes$ indicates the channel-wise multiplication and $\mathbf{w}_j \in \mathbb{R}^C$ is the weights. The $\otimes$ is defined as: for a vector $\mathbf{v} = [v_1 \ v_2 \ \cdots \ v_C]$ and a tensor $\mathbf{M} = [\mathbf{m}_1 \ \mathbf{m}_2 \ \cdots \ \mathbf{m}_C]$ with $C$ feature channels, $\mathbf{v} \otimes \mathbf{M} = [v_1 * \mathbf{m}_1 \ v_2 * \mathbf{m}_2 \ \cdots v_C * \mathbf{m}_C]$.

We implement the temporal aggregation as a network module (Fig. 2). It involves three steps as follows,

1. apply $1 \times 1$ depthwise convolution $r$ times to $n$ input tensors to form an output matrix of size $r \times n$;

2. shift the $i^{th}$ row left (or right) by $|i - \lfloor r/2 \rfloor|$ positions if $i > \lfloor r/2 \rfloor$ (or $i \le \lfloor r/2 \rfloor$) and if needed, pad leading or trailing zero tensors in the front or at the end;

3. perform temporal aggregation along the column to generate the output.

The aggregation module(*TAM*), highlighted as a red box in Fig. 1d), is inserted as a separate layer after the local temporal fusion in the *bLVNet*, resulting in the final *bLVNet-TAM* architecture. Obviously none of the steps in the implementation above involve costly computation, so the module is fairly fast. A node in the network initially only sees $r - 1$ neighbors. As the network goes deeper, the amount of context that the node involves in the input grows quickly, similar to how the receptive field of a neuron is enlarged in a CNN. In such a manner, long-range temporal dependencies are thus potentially captured. For this reason, the temporal aggregation is also called *global temporal fusion* here, as opposed to the *local temporal fusion* discussed above.

The work of TSM [4] has also applied temporal shifting to swap feature channels between neighboring frames. In such a case, TSM can be treated as a special case of our method where the weights are empirically set rather than learned from data. In Section 4.3, we demonstrate that the proposed TAM is more effective than TSM for temporal modeling under different video architectures. TAM is also related to S3D [15] and R(2+1)D [16] in that TAM is independent of spatial convolutions. However, TAM is based on depthwise convolution, thus has fewer parameters and less computation than S3D and R(2+1)D.

The *TAM* can also be integrated into 3D convolutions such as C3D [20] and I3D [1] to further enhance the temporal modeling capability that already exists in these models. Due to the difference in how temporal data is presented between 2D-based and 3D-based models, the temporal shifting now needs to operate on feature channels within a tensor instead of on tensors themselves.

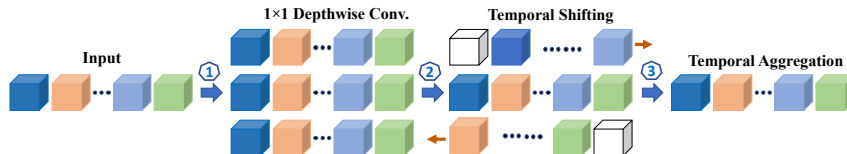

Figure 2: Temporal aggregation module (TAM). The TAM takes as input a batch of tensors, each of which is the activation of a frame, and produces a batch of tensors with the same order and dimension. The module consists of three operations: 1) $1 \times 1$ depthwise convolutions to learn a weight for each feature channel; 2) temporal shifts (left or right direction indicated by the smaller arrows; the white cubes are padded zero tensors.); and 3) aggregation by summing up the weighted activations from 1).

# 4 Experiments

## 4.1 Experimental Setup

**Datasets**. We evaluate our approach on three large-scale datasets for video recognition, including the widely used Something-Something (Version 1 and Version 2) [9], Kinetics-400 [6] and the recent Moments-in-time dataset [10]. They are herein referred to as *SS-V1*, *SS-V2*, *Kinetics-400* and *Moments*, respectively.

Something-Something is a dataset containing videos of 174 types of predefined human-object interactions with everyday objects. The version 1 and 2 include 108k and 220k videos, respectively. This dataset focuses on human-object interactions in a rather simple setup with no scene contexts to be exploited for recognition. Instead temporal relationships are as important as appearance for reasoning about the interactions. Because of this, the dataset serves as a good benchmark for evaluating the efficacy of temporal modeling, such as proposed in our approach.

Kinetics-400 [6] has emerged as a standard benchmark for action recognition after UCF101 [21] and HMDB [22], but on a significantly larger scale. The dataset consists of 240k training videos and 20k validation videos, with each video trimmed to around 10 seconds. It has a total of 400 human action categories.

Moments-in-time [10] is a recent collection of one million labeled videos, involving actions from people, animals, objects or natural phenomena. It has 339 classes and each video clip is trimmed to 3 seconds long.

**Data Augmentation**. During training, we follow the data augmentation used in TSN [5] to augment the video with different sizes spatially and flip the video horizontally with 50% probability. Furthermore, since our models are finetuned on pretrained ImageNet, we normalize the data with the mean and standard deviation of the ImageNet images. The model input is formed by *uniform sampling*, which first divides a video into $n$ uniform segments and then selects one random frame from each segment as the input.

During inference, we resize the smaller side of an image to 256 and then crop a centered $224 \times 224$ region. The center frame of each segment in uniform sampling is picked as the input. On Something-Something and Moments, our results are based on the single-crop and single-clip setting. On *Kinetics-400*, we use the common practice of multi-crop and multi-clip for evaluation.

**Training Details**. Since all the three datasets are large-scale, we train the models in a progressive way. For each type of backbone (for example, *bLResNet*-50), we first finetune a base model on ImageNet with a minimum input length (i.e. $8 \times 2$ in our case) using 50 epochs. We adopt the Nesterov momentum optimizer with an initial weight of 0.01, a weight decay of 0.0005 and a momentum of 0.9. We then finetune a new model with longer input (for example, $16 \times 2$) on top of the corresponding base model, but with 25 epochs only. In this case, the initial learning rate is set to 0.01 on Something-Something and 0.005 on Kinetics and Moments. The learning rate is decreased by a factor of 10 at the 10-th and 20-th epoch, respectively.

This strategy allows to significantly reduce the training time needed for all the models evaluated in our experiments. All our models were trained on a server with 8 GPU cards and a total of 128G GPU memory. We set the total batch size to 64 whenever possible. For models that require more memory to train, we adjust the batch size accordingly to the maximum number allowed.

## 4.2 Main Results

**Something-Something**. We first report our results on the validation set of the Something-Something datasets in Table 1 and Table 2. With a moderately deep backbone *bLResNet-50*, our approach outperforms all 3D models on *SS-V1* while using much fewer input frames ($8 \times 2$) and being substantially more efficient. TSM [4] was the previously best approach on Something-Something. Under the same backbone (i.e. ResNet-50), our approach is better than TSM on both *SS-V1* and *SS-V2* while being more efficient (i.e our 8x2 model has $1.4$ times fewer FLOPs than a 8-frame TSM model).

When empowered with a stronger backbone *bLResNet-101*, our approach achieves even better results at $32 \times 2$ frames (**53.1%** top-1 accuracy on *SS-V1*, and **65.2%** on *SS-V2*), establishing a new state-of-the-art on Something-Something. Notably, these results while based on RGB information only,

Table 1: Recognition Accuracy of Various Models on Something-Something-V1 (SS-V1).

| Model | Backbone | Pretrain | Frames | Modality | Param ($10^6$) | FLOPs ($10^9$) | Val Top-1 (%) | Val Top-5 (%) | Test Top-1 (%) |
|---|---|---|---|---|---|---|---|---|---|
| I3D [1] | Inception | ImageNet | 64 | RGB | 12.7 | 111 | 45.8 | 76.5 | 27.2 |
| NL I3D + GCN [23] | ResNet-50 | ImageNet | 32+32 | RGB | 303 | 62.2 | 46.1 | 76.8 | – |
| S3D [15] | Inception | ImageNet | 64 | RGB | 8.77 | 66 | 47.3 | 78.1 | – |
| ECO-Lite$_{En}$ [24] | BNInception+ResNet18 | ImageNet | 92 | RGB | 150 | 267 | 46.4 | – | 42.3 |
| TSN [5] | BNInception | ImageNet | 8 | RGB | 10.7 | 16 | 19.5 | – | – |
| TRN [3] | BNInception | ImageNet | 8 | RGB | 18.3 | 16 | 34.4 | – | 33.6 |
| | BNInception | ImageNet | 8+8 | RGB+Flow | – | – | 42.0 | – | 40.7 |
| TSM [4] | ResNet-50 | Kinetics | 8 | RGB | 24.3 | 33 | 45.6 | 74.2 | |
| | ResNet-50 | Kinetics | 16 | RGB | 24.3 | 65 | 47.2 | 77.1 | 46.0 |
| | ResNet-50 | Kinetics | 16+16 | RGB+Flow | – | – | 52.6 | 81.9 | 50.7 |
| bLVNet-TAM | bLResNet-50 | ImageNet | 8×2 | RGB | 25.0 | 23.8 | 46.4 | 76.6 | – |
| | bLResNet-50 | SS-V1 | 16×2 | RGB | 25.0 | 47.7 | 48.4 | 78.8 | – |
| | bLResNet-101 | ImageNet | 8×2 | RGB | 40.2 | 32.1 | 47.8 | 78.0 | – |
| | bLResNet-101 | SS-V1 | 16×2 | RGB | 40.2 | 64.3 | 49.6 | 79.8 | – |
| | bLResNet-101 | SS-V1 | 24×2 | RGB | 40.2 | 96.4 | 52.2 | 81.8 | – |
| | bLResNet-101 | SS-V1 | 32×2 | RGB | 40.2 | 128.6 | **53.1** | **82.9** | **48.9** |

Table 2: Recognition Accuracy of Various Models on Something-Something-V2 (SS-V2).

| Model | Backbone | Pretrain | Frames | Modality | Param ($10^6$) | FLOPs ($10^9$) | Val Top-1 (%) | Val Top-5 (%) | Test Top-1 (%) | Test Top-5 (%) |
|---|---|---|---|---|---|---|---|---|---|---|
| TRN [3] | BNInception | ImageNet | 8 | RGB | 18.3 | 16 | 48.8 | 77.6 | 50.9 | 79.3 |
| | BNInception | ImageNet | 8 | RGB+Flow | 36.6 | 32 | 55.5 | 83.1 | 56.2 | 83.2 |
| TSM [4] | ResNet-50† | Kinetics | 8 | RGB | 24.3 | 33 | 58.9 | 85.5 | – | – |
| | ResNet-50† | Kinetics | 16 | RGB | 24.3 | 65 | 61.4 | 87.0 | – | – |
| | ResNet-50 | Kinetics | – | RGB+Flow | – | – | 66.0 | 90.5 | 66.6 | 91.3 |
| bLVNet-TAM | bLResNet-50 | ImageNet | 8×2 | RGB | 25.0 | 23.8 | 59.1 | 86.0 | – | – |
| | bLResNet-50 | SS-V2 | 16×2 | RGB | 25.0 | 47.7 | 61.7 | 88.1 | – | – |
| | bLResNet-101 | ImageNet | 8×2 | RGB | 40.2 | 32.1 | 60.2 | 87.1 | – | – |
| | bLResNet-101 | SS-V2 | 16×2 | RGB | 40.2 | 64.3 | 61.9 | 88.4 | – | – |
| | bLResNet-101 | SS-V2 | 24×2 | RGB | 40.2 | 96.4 | 64.0 | 89.8 | – | – |
| | bLResNet-101 | SS-V2 | 32×2 | RGB | 40.2 | 128.6 | 65.2 | 90.3 | – | – |
| | bLResNet-101* | SS-V2 | 32×2 | RGB+Flow | – | – | 68.5 | 91.4 | **67.1** | **91.4** |

† : using their pretrained models and code to evaluate under the 1-crop and 1-clip setting for fair comparison
* : model ensemble of RGB and Flow model, each is evaluated with 3 crops and 10 clips and uses 256 as the shorter side.

are superior to those obtained from the best two-stream models at no more computational cost. This strongly demonstrates the effectiveness of our approach for temporal modeling. We further evaluated our models on the test set of Something-Something. Our results are consistently better than the best results reported by the other approaches in comparison including 2-stream models.

**Kinetics-400**. Kinetics-400 is one of the most popular benchmarks for action recognition. Currently the best-performed models on this dataset are all based on 3D Convolutions. However, it has been shown in the literature that temporal ordering in this dataset does not seem to be as crucial as RGB information for recognition. For example, as experimented in S3D [15], the model trained on normal time-order data performs well on the time-reversed data on Kinetics. In accordance to this, our approach (3 crops and 3 clips) mainly performs on par with or better than the current large-scale architectures, but without outperforming them as clearly as on the Something-Something datasets, where the temporal relations are more essential for an overall understanding of the video content.

Table 3: Recognition Accuracy of Various Models on Kinetics-400 (RGB-only).

| Net | Backbone | Pretrain | FLOPs ($10^9$) | Top-1 (%) | Top-5 (%) |
|---|---|---|---|---|---|
| STC [25] | ResNeXt-101 | None | – | 68.7 | 88.5 |
| ARTNet [26] | ResNet-18 | None | 23.5×250 | 69.2 | 88.3 |
| C3D [26] | ResNet-18 | None | 19.6×250 | 65.6 | 85.7 |
| I3D [1] | Inception | ImageNet | 108×N/A | 71.1 | 89.3 |
| S3D [15] | Inception | ImageNet | – | 72.2 | 90.6 |
| R(2+1)D [16] | ResNet-34 | None | – | 72.0 | 90.0 |
| SlowFast-4×16 [17] | ResNet-50 | None | 36.1×30 | **75.6** | **92.1** |
| TSN [5] | InceptionV3 | ImageNet | 142.8×10 | 72.5 | – |
| ECO-Lite$_{En}$ [24] | BNInception+ResNet18 | ImageNet | 267 | 70.7 | - |
| TSM-8 [4] | ResNet-50 | ImageNet | 42.7×30 | 74.1 | 91.2 |
| TSM-16 [4] | ResNet-50 | ImageNet | 85.4×30 | 74.7 | – |
| bLVNet-TAM-8×2 | bLResNet-50 | ImageNet | 31.1×9 | 71.0 | 89.8 |
| bLVNet-TAM-16×2 | bLResNet-50 | Kinetics | 62.3×9 | 72.0 | 90.6 |
| bLVNet-TAM-24×2 | bLResNet-50 | Kinetics | 93.4×9 | 73.5 | 91.2 |

Table 4: Recognition Accuracy of Various Models on Moments-in-time.

| Net | Backbone | Pretrain | Frames | Modality | Top-1 (%) | Top-5 (%) |
|---|---|---|---|---|---|---|
| SoundNet [10] | — | — | — | Audio | 7.60 | 18.0 |
| TSN [10] | BNInception | ImageNet | 16 | RGB | 24.1 | 49.1 |
| TSN [10] | BNInception | — | 16+16 | RGB+Flow | 25.3 | 50.1 |
| TRN [10] | Inception | ImageNet | 16 | RGB | 28.3 | 53.9 |
| I3D [10] | ResNet-50 | — | 16 | RGB | 29.5 | 56.1 |
| Ensemble [10] | — | — | — | — | 31.2 | 57.7 |
| *bLVNet-TAM* | *bLResNet*-50 | ImageNet | 8×2 | RGB | 31.2 | 58.3 |
| | *bLResNet*-50 | Moments | 16×2 | RGB | **31.4** | **59.3** |

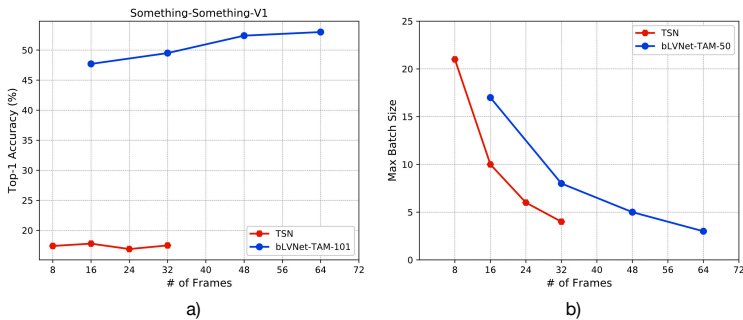

a)         b)

Figure 3: Number of input frames v.s. model accuracy and memory usage. (a) A longer input sequence yields better recognition in our proposed *bLVNet-TAM* on the Something-Something dataset [9], but not in TSN [5] due to limited temporal modeling ability. (b) Compared to TSN, *bLVNet-TAM* reduces memory usage by ∼2 times under the same number of input frames.

**Moments**. We finally evaluate the proposed architecture on the *Moments* dataset [10], a large-scale action dataset with about three times more training samples than *Kinetics-400*. Since *Moments* is relatively new and results reported on it are limited, we only compare our results with those reported in the Moments paper [10]. As can been seen from Table 4, our approach outperforms all the single-stream models as well as the ensemble one. We hope our models provide stronger baseline results for future reference on this challenging dataset.

It is also noted that our model trained with $16 \times 2$ frames only produces slightly better top-1 accuracy than the model trained with $8 \times 2$ frames. We speculate that this has to do with the fact that the *Moments* clips are only as short as 3 seconds and that there is only a limited impact in choosing a finer temporal granularity on this dataset.

## 4.3 Ablation Studies

In this section, we conduct ablation studies to provide more insights about our main ideas.

**Is temporal aggregation effective?**. We validate the efficacy of the proposed temporal aggregation module (*TAM*), which is considered as a global fusion method (Section 3.3). *Local fusion* here is referred to the branch merging in the dual path network (Section 3.2). We compare *TAM* with the temporal shift module used in TSM [4] in Table 5 under two different video architectures: TSN and *bLVNet* proposed in this work. *TAM* demonstrates clear advantages over TSM, outperforming TSM by over 2% under both architectures. Interestingly, with the here proposed *bLVNet* baseline with local temporal fusion almost doubles the performance of a TSN baseline, improving the accuracy from 17.4% to 33.6%. On top of that, *TAM* boosts the performance by another 13% in both cases, suggesting that *TAM* is complementary to local fusion. This further confirms the significance of temporal reasoning on the Something-Something dataset.

**Does seeing more frames help?**. One of the main contribution of this work is an efficient video architecture that makes it possible to train deeper models with more input frames using moderate GPU resources. Fig. 3a) shows consistent improvement of our approach on *SS-V1* as the number of input frames increases. A similar trend in our results can be observed on *Kinetics-400* in Table 3. On the other hand, the almost flattened line from TSN suggests that a model without effective temporal modeling cannot take much of the benefit from longer input frames.

**Memory Usage**. We compare the memory usage between our approach based on *bLResNet*-50 and TSN based on ResNet-50. As shown in Fig. 3b), our approach is more memory friendly than TSN, achieving a saving of $\sim$2 times at the same number of input frames. The larger batch size allowed for training under the same computational budget is critical for our approach to obtain better models and reduce training time.

## 5  Conclusion

We presented an efficient and memory-friendly video architecture for learning video representations. The proposed architecture allows for twice as many input frames as the baseline while using less computation and memory. This enables training of deeper models with richer input under the same GPU resources. We further developed a temporal aggregation method to capture temporal dependencies effectively across frames. Our models achieve strong performance on several action recognition benchmarks, and establish a state-of-the-art on the Something-Something dataset.

Table 5: Temporal Modeling on SS-V1.

| Net | Backbone | Local Fusion | Global Fusion | Top-1 (%) |
|---|---|---|---|---|
| TSN | ResNet-50 | None | None | 17.4 |
| | ResNet-50 | None | TSM | 43.4 |
| | ResNet-50 | None | TAM | 46.1 |
| bLVNet | *bLResNet*-50 | ✓ | None | 33.6 |
| | *bLResNet*-50 | ✓ | TSM | 44.2 |
| | *bLResNet*-50 | ✓ | TAM | **46.4** |

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
