[Supplementary Material]

## A  Network Architecture

Here we details our network architecture for *bLVNet-TAM*-50 in Table 6. We follow the notation used in *bLNet* [7] ($\alpha = 2$ and $\beta = 4$) but adding the proposed TAM module before branching out to *Big-Net* and *Little-Net* and the last shared residual block. As noted before, the two branches work on different frames and then merged every stage; on the other hand, in ResBlock$_{TAM}$, the TAM module goes through the non-shortcut path.

Table 6: Network configurations of *bLVNet-TAM*-50 with temporal fusion.

| Layers | Spatial output size | *bLVNet-TAM*-50 | |
|---|---|---|---|
| Convolution | $112 \times 112$ | $7 \times 7, 64, s2$ | |
| TAM-module | $112 \times 112$ | Temporal Aggregation Module ($r = 3$) | |
| bL-module | $56 \times 56$ | $3 \times 3, 64, s2$ | $\begin{pmatrix} 3{\times}3, 32 \\ 3{\times}3, 32, s2 \\ 1{\times}1, 64 \end{pmatrix}$ |
| TAM-module | $56 \times 56$ | Temporal Aggregation Module ($r = 3$) | |
| bL-module | $56 \times 56$ / $28 \times 28$ | ResBlock$_B$, 256 $\times 2$　　　ResBlock$_L$, 128 $\times 1$ <br> ResBlock, 256, s2 | |
| TAM-module | $28 \times 28$ | Temporal Aggregation Module ($r = 3$) | |
| bL-module | $28 \times 28$ / $14 \times 14$ | ResBlock$_B$, 512 $\times 3$　　　ResBlock$_L$, 256 $\times 1$ <br> ResBlock, 512, s2 | |
| TAM-module | $14 \times 14$ | Temporal Aggregation Module ($r = 3$) | |
| bL-module | $14 \times 14$ / $14 \times 14$ | ResBlock$_B$, 1024 $\times 5$　　　ResBlock$_L$, 512 $\times 1$ <br> ResBlock, 1024 | |
| ResBlock$_{TAM}$ | $7 \times 7$ | ResBlock$_{TAM}$, 2048 $\times 3$, s2 | |
| Average pool | $1 \times 1$ | $7 \times 7$ average pooling | |
| FC, softmax | | # of classes | |

ResBlock$_B$: the first $3 \times 3$ convolution is with stride 2, and then restoring the size via the bi-linear upsampling.
ResBlock$_L$: a $1 \times 1$ convolution is applied at the end to align the channel size.
ResBlock$_{TAM}$: a residual block embedded with temporal aggregation module with $r = 3$.
s2: the stride is set to 2 for the $3 \times 3$ convolution in the ResBlock.

## B  Data Preprocessing

Here we describe how we convert the video data into images for our training and inference. For the Something-Something dataset, we resize the smaller side of an image to 256 while keeping aspect ratio. For the Kinetics dateset, we resize the smaller side of an image to 331 since its original resolution is higher. For the Moments dataset, we we resize an image to 256×256.