[Reviews · NeurIPS 2019]

Reviewer 1



+ Three challenging and large-scale video datasets are used: Something-Something, Kinetics, Moments-in-time. + Various ablations are provided. The experimental analyses lead to very interesting observations that are useful for the community. The fact that TSN [15] does not improve with more #frames, whereas bLVNet-TAM does is interesting to see (Fig 3). The performance breakdown for different components in Table 5 is nice. Overall, there are many take-home messages to learn from this paper’s experiments. - In terms of methodology, the paper combines existing blocks: Big-Little-Net [7], TSN [15], TSM [4]. Therefore, its technical contribution is limited. * Final rating: I keep my initial score, i.e. 6, after having read the rebuttal and the other reviews. I thank the authors for the clarifications provided. Despite the limited originality, I believe that the paper can be a valuable contribution to the community with its simplicity, positive results, and comprehensive experiments. I encourage the authors to incorporate the clarifications in the revised version.

Reviewer 2



Strengths: * Simplicity. Both the bLVNet and the TAM are simple models, easy to implement and probably fairly straightforward to train. This is a good property. * Paper is well-written and the technical approach is easy to comprehend. * Although the TAM is demonstrated using a frame-based 2D CNN, it is straightforward to extend to 3D CNNs, with potential further gains in accuracy. * Comprehensive evaluation on 3 large-scale video datasets shows the memory/efficiency/accuracy gains enabled by the two proposed schemes (bLVNet and TAM). Weaknesses: * Technical innovation is fairly limited. The bLVNet is a straightforward extension of bLNet (an image model) to video. The TAM involves the use of 1D temporal convolution and depthwise convolution. Both mechanisms that have been widely leveraged before. On the other hand, the paper does not make bold novelty claims and recognizes the contribution as being more empirical than technical. The TAM shares many similarities with Timeception [Hussein et al., CVPR 19], which was not yet published at the time of this submission and thus does not diminish the value of this work. Nevertheless, given the many analogies between these concurrent approaches, it'd be advisable to discuss their relations in future versions (or the camera-ready version) of the paper. * While the memory/efficiency gains are convincingly demonstrated, they are not substantial enough to be a game-changer in the practice of training video understanding models. Due to the overhead of setting up the proposed framework (even though quite simple), adoption by the community may be fairly limited. Final rating: - After having read the other reviews and the author responses, I decide to maintain my initial rating (6). The contribution of this work is mostly empirical. The stronger results compared to more complex models and the promise to release the code imply that this work deserves to be known, even if fairly incremental.

Reviewer 3



1. Originality: WEAK a) I think this is the weakest part of this paper. Almost all of the contributions of this work have been individually explored. For example: b) The idea of parallel pathways for low-res/high-res processing was explored in SlowFast, 2-stream networks etc. Granted, authors use a somewhat different design from the ICLR'19 paper, where the spatial resolution is changed across the pathways, but the core idea is fairly well explored. c) The aggregation layer (TAM) is essentially TSM [4] with learned weights, and gets a few percentage points extra in performance. d) Missing related work: Aggregating context temporally for video representation learning has been explored in many previous works, which would be good to report in the related work. I point some here. - Aggregating using VLAD/Fisher vectors etc: Action recognition with stacked fisher vectors (ECCV'14), Learnable pooling with Context Gating for video classification (CVPR'17), ActionVLAD (CVPR'17), SeqVLAD (TIP'18) etc - Aggregation using attention: Attention Clusters (CVPR'18), Video Action Transformer Network (CVPR'19), Long-Term Feature Banks for Detailed Video Understanding (CVPR'19) etc - Other temporal modeling architectures: Timeception for Complex Action Recognition (CVPR'19), Videos as space-time region graphs (ECCV'18) etc 2. Quality: GOOD I think authors do a good job of doing thorough experiments, and comparing performance of recent works along with computational/memory costs. The ablations are useful as well. 3. Clarity: WEAK Quite a few aspects of the model were not immediately clear to me. I would encourage authors to clarify in the rebuttal: a) Splitting video into odd/even frames, setting odd as big and even as little: This seems very adhoc. Why enforce this rule? Why not just use pairs of frames and use one at lower resolution and other at higher? Is there a reason odd frames in the video must be bigger? b) What is the train-time complexity of the model? Since the aggregation layer has to be trained, and needs at least "r" clips temporally-shifted clips at the same time, it would limit the training batch size (something like TSN would not have that issue). Is that a limiting factor at all? I would like to see more discussion on that aspect in the final version. c) L221: What is "single-crop single-frame" testing? I assume it is done in TSN style -- so for SS-V1 model which uses 32x2 frames, you have 32 segments at test time and use a pair of frames from each segment (odd and even). d) If my understanding in (c) is correct, then what is the "multi-crop" setup used in Kinetics? How many frames are being used in Table 3? e) I am assuming the "Frames" column in the tables reports the *TOTAL* frames used in inference, including multiple crops etc. Is that correct? 4. Significance: MODERATE While the work doesn't significantly improve on the state of the art, it does seem to propose a cheaper alternative. That can be very useful for research groups with limited resources to work on related areas, if the code is made available. However it's not clear from the paper if the code for reproducing the reported results be released? Final rating ======== I have looked through the other reviews and author feedback. I appreciate authors efforts in responding to my concerns, and clarifying parts of the paper. As all reviewers note, the technical novelty of the work is limited, though the good performance on standard benchmarks with lower computation might be valuable. Given the newer results in rebuttal and the promise to release code, I am upgrading my rating to 6. However, I still think the writing and presentation at least needs quite a bit more work to explain their approach and setup clearly.

[Author Response · NeurIPS 2019]

We thank all the reviewers for their constructive comments. Please see our responses below.

**R1-3: novelty and relevant work.** The key contribution of our work is the development of an efficient and memory-
friendly architecture for video understanding. Our approach is purely based on 2D convolutions. Nevertheless, it
outperforms or performs comparably to many more costly 3D models. Especially, our proposed TAM layers have
been shown more effective than 3D temporal convolutions and some recently proposed spatiotemporal approaches
that are structurally more sophisticated (Table 1 and 3). We hope that our findings and results in the paper are helpful
and will make the community to rethink the efficacy of 2D and 3D architectures in learning spatiotemporal feature
representations.

We thank the reviewers for pointing out some related (or missing) references. We note that some of them such as
Timeception, SlowFast and TSM are concurrent with our work. Here we briefly describe the main differences between
these approaches and ours, and more discussions will be added to the final manuscript. Timeception basically applies
the concept of "Inception" to the temporal domain for capturing long-range temporal dependencies in a video. The
Timeception layers involve group convolutions at different time scales while our TAM layers only use depthwise
convolution. As a result, the Timeception has significantly more parameters than the TAM (10% vs. 0.1% of the
total model parameters). As for SlowFast, it differs from our approach in that a) it uses 3D convolutions for temporal
modeling; and b) it achieves efficiency by balancing the number of input frames and channels at different network
branches. Compared to TSM, our approach is more generalized, more extensible, and in particular more effective, as
shown in the paper.

**R1-3: new results.** After the submission, we further trained optical-flow models on the Something-Something V2
dataset and applied model ensemble with the corresponding RGB models. Our 2-stream models improve top-1 accuracy
over the RGB models by **2.2%-2.8%** on the validation set. On the leaderboard, we are currently the $2^{nd}$ best on top-1
accuracy and the $1^{st}$ on top-5 accuracy.

**R2 and R3: code release.** We will release our code and models for this work as well as the scripts for data preparation,
model training and evaluation. In the meanwhile, we would be delighted to share as much material as possible to
help independent replication and validation of our work. In addition, the Big-Little Net code is publicly available at
`https://github.com/IBM/BigLittleNet`, which should be helpful for the adoption of our work.

**R1: performance of 8×2 models on ImageNet.** We realized that Line 255 in the paper might have confused R1.
To clarify that, all our models using 8×2 frames in Table 1-4 (*bLVNet-TAM*-8 × 2) were fine tuned from 2D models
pretrained on ImageNet. Then, the models *bLVNet-TAM*-16 × 2 were learnt from *bLVNet-TAM*-8 × 2 and *bLVNet-TAM*-
24 × 2 from *bLVNet-TAM*-16 × 2 and *bLVNet-TAM*-32 × 2 from *bLVNet-TAM*-24 × 2, respectively. We found that
learning in such a progressive way is not only effective, but also faster than fine tuning from ImageNet.

**R3: odd or even frames as input.** We do not enforce that the big branch must operate on odd frames and the little
branch on even frames. Instead the big branch can take either of a pair of frames as input and the other frame goes to
the little branch. We will clarify this in the final manuscript.

**R3: complexity of TAM.** Like TSN, our approach has a training-time complexity proportional to the number of input
frames because the TAM layers are highly light-weighted compared to the backbone network. As shown in Fig. 3 in the
paper, when using the same number of input frames, our models allow for a batch size of about 2 times larger than TSN
in training. Note that The TAM operates on 'r' *frames* rather than 'r' *clips*.

**R3: evaluation setup (question c-e)).** R3 is right about the "single-crop single-clip" setup, which means a single clip
is formed for each video in test by picking a pair of frames from a set of uniformly split segments of the video. The
results in Table 1, 2 and 4 are reported based on such a setup. The 'Frames' column refers to the total number of
frames used in inference, but with a single crop per frame only. Differently, the "multi-crop multi-clip" setup can be
considered as repeating "single-crop single-clip" multiple times at different time instances and at different cropping
locations in a test video. In such a case, the TOTAL number of frames used in inference is thus the product of the
number of frames used in "single-crop single-clip", the number of crops and the number of clips. For example, in Table
3, our *bLVNet-TAM*-8×2 uses 16×3 (crops)×3 (clips) frames for evaluation while TSM-8 uses 8×3 (crops)×10 (clips)
frames.

**R3: performance improvement over SOTA.** While being efficient, our approach achieves the state of the art accuracy
on the Something-Something and Moments datasets (see Table 1, 2 and 4). It's worthy to note that our approach
only uses RGB information, but still outperforming the previously best 2-stream models based on both RGB and
optical flow information. In addition, our recently trained two-stream model (*bLVNet-TAM*-32×2) is 2.8% better than
TSM-16 at top-1 accuracy (66.8% vs. 64.0%) on the SS-V2 validation set, and our approach are ranked the $2^{nd}$ on the
Something-Something leaderboard (2% better than the TSM-16 on the leaderboard, 66.34% vs. 64.33%).

[Meta-Review · NeurIPS 2019]

This is a borderline case. The initial scores for this paper were: 6: Marginally above the acceptance threshold. 6: Marginally above the acceptance threshold. 5: Marginally below the acceptance threshold. Positive points: + extensive experimental analysis on three large-scale video datasets (Something-Something, Kinetics, Moments-in-time) showing memory/efficiency/accuracy gains enabled by the proposed approach. + several ablations providing insights likely to be useful for the community. + simplicity of the method + clear writing Negative points: - the paper combines together existing building blocks (Big-Little-Net, TSN, TSM) and hence has somewhat limited novelty. - “memory/efficiency gains are convincingly demonstrated, but are not substantial enough to be a game-changer in the practice”. - Missing important details. The authors provide a rebuttal. After seeing the rebuttal and in the follow-up discussion R1 and R2 maintain their slightly positive rating (6) and R3 upgrades their rating from 5 to 6. The rebuttal addresses some of the concerns, though the concern regarding limited novelty remains. All reviewers have also updated their review with post-rebuttal comments: R1: “Despite the limited originality, I believe that the paper can be a valuable contribution to the community with its simplicity, positive results, and comprehensive experiments.” R2: “The contribution of this work is mostly empirical. The stronger results compared to more complex models and the promise to release the code imply that this work deserves to be known, even if fairly incremental.” R3: “Technical novelty of the work is limited, though the good performance on standard benchmarks with lower computation might be valuable. Given the newer results in rebuttal and the promise to release code, I am upgrading my rating to 6.” AC is convinced by the positive arguments of the reviewers and recommends accept. The authors are strongly encouraged to incorporate the new results from the rebuttal and the clarifications suggested by the reviewers into the camera ready version of the paper.